# The Body Fat Percentage Rather Than the BMI Is Associated with the CD4 Count among HIV Positive Japanese Individuals

**DOI:** 10.3390/nu14030428

**Published:** 2022-01-18

**Authors:** Kumiko Shoji, Michinori Shirano, Mitsuru Konishi, Yuko Toyoshima, Miyuki Matsumoto, Tetsushi Goto, Yu Kasamatsu, Yuji Ichida, Yasuo Kagawa, Terue Kawabata, Hiromitsu Ogata, Daiki Habu

**Affiliations:** 1Faculty of Nutrition, Kagawa Nutrition University, Saitama 350-0288, Japan; kagawa@eiyo.ac.jp (Y.K.); kawabata@eiyo.ac.jp (T.K.); ogata.hiromitsu@eiyo.ac.jp (H.O.); 2Department of Infectious Disease, Osaka City General Hospital, Osaka 534-0021, Japan; shirano@wonder.ocn.ne.jp (M.S.); gototet@yahoo.co.jp (T.G.); hmorimoto2002@yahoo.co.jp (Y.K.); 3Center for Health Control, Nara Medical University, Nara 634-8521, Japan; mkonishi@naramed-u.ac.jp; 4Department of Nursing, Osaka City General Hospital, Osaka 534-0021, Japan; yu-toyoshima@osakacity-hp.or.jp; 5Kurokawa Clinic, Osaka 557-0031, Japan; ayasuke61720@cap.ocn.ne.jp; 6Department of Infectious Disease, University Hospital, Kyoto Prefectural University of Medicine, Kyoto 602-8566, Japan; 7Department of Pharmacy, Osaka City General Hospital, Osaka 534-0021, Japan; yk.ichida@space.ocn.ne.jp; 8Graduate School of Human Life Science, Osaka City University, Osaka 558-8585, Japan; habu@osaka-cu.ac.jp

**Keywords:** HIV, CD4 count, body fat percentage, cardiovascular disease risk

## Abstract

Maintenance of the cluster of differentiation 4 (CD4) positive lymphocyte count (CD4 count) is important for human immunodeficiency virus (HIV) positive individuals. Although a higher body mass index (BMI) is shown to be associated with a higher CD4 count, BMI itself does not reflect body composition. Therefore, we examined the association of body weight, body composition and the CD4 count, and determined the optimal ranges of CD4 count associated factors in Japanese HIV positive individuals. This cross-sectional study included 338 male patients treated with antiretroviral therapy for ≥12 months. Multiple logistic regression analysis was used to identify factors significantly associated with a CD4 count of ≥500 cells (mm^3^)^−1^. The cutoff values of factors for a CD4 ≥ 500 cells (mm^3^)^−1^ and cardiovascular disease risk were obtained by receiver operating characteristic curves. Age, body fat percentage (BF%), nadir CD4 count, duration of antiretroviral therapy (ART), years since the HIV-positive diagnosis and cholesterol intake showed significant associations with the CD4 count. The cutoff value of BF% for a CD4 ≥ 500 cells (mm^3^)^−1^ and lower cardiovascular disease risk were ≥25.1% and ≤25.5%, respectively. The BF%, but not the BMI, was associated with CD4 count. For the management of HIV positive individuals, 25% appears to be the optimal BF% when considering the balance between CD4 count management and cardiovascular disease risk.

## 1. Introduction

With the development of antiretroviral therapy (ART), the mortality rate of human immunodeficiency virus (HIV)-positive individuals has drastically improved [1]. In the absence of risk factors, such as comorbid viral hepatitis, drug abuse, and smoking, the life expectancy of HIV-positive individuals is very close to that of HIV-negative individuals [2,3]. It was also shown that the mortality rate did not differ significantly between healthy individuals and HIV-positive men with a CD4 positive lymphocyte count (CD4 count) of 500/mm^3^ or higher and no risk factors [4]. This indicates that in addition to the HIV RNA levels, which can be controlled relatively well with ART, the control of CD4 count is important for HIV positive individuals.

Recently, there was an increasing number of reports showing that weight and body fat (BF) increase with ART, particularly with integrase strand transfer inhibitors (INSTIs) [5,6,7]. In addition, in areas of the world with abundant food resources and a good health care system, the rates of overweight and obesity are higher than the rates of undernutrition, which was the most important issue before the introduction of ART [8]. There are several reports showing a relationship between body weight and the CD4 count, e.g., a 1-unit increase in the body mass index (BMI) raised the CD4 count by 8.7 cells (mm^3^)^−1^ [9], and a higher BMI was associated with long-term advantages in CD4 count recovery with ART [10]. However, the BMI itself does not reflect body composition [11,12]; it remains unclear whether it is the BF, muscle or body weight that is associated with the CD4 count.

This study examined the associations among body weight, body composition and the CD4 count among Japanese men living with HIV, taking into consideration their nutritional intakes and other lifestyle factors. At the same time, the optimal ranges of the factors associated with the CD4 count were determined for consideration of the balance between CD4 count management and the risk of cardiovascular disease.

## 2. Materials and Methods

### 2.1. Study Design and Subjects

This was a cross-sectional study conducted at two hospitals in Japan between 2014 and 2017. A total of 338 patients were included. Since it was reported that the growth of HIV can be suppressed after 1 year of ART [13], this study was limited to those who had been on ART for at least 12 months. Figure 1 shows the flowchart for subject selection, and 289 patients were included for the final analysis.

### 2.2. Anthropometric Measurement

All of the anthropometric indices were measured in each patient during one of their outpatient visits. The body weight, BF, muscle mass and BF percentage (BF%) were measured using a bioimpedance method by InBody230 (InBody Japan Inc., Tokyo, Japan).

### 2.3. Dietary Intake

The energy intake, nutritional intake, and the intake of individual food groups were calculated from the frequency of intake of 58 foods commonly consumed by Japanese people using the brief self-administered diet history questionnaire (BDHQ) developed by Sasaki et al. [14,15]. Responses to the questionnaire were requested on the same day of anthropometric measurement. The exclusion criteria for reporting errors on BDHQ were values < 0.5 times the energy requirement of Physical Activity Level I or >1.5 times the energy requirement of Physical Activity Level III for each subject. Energy adjustments were made based on the density method (nutrient intake per 1000 kcal of energy intake).

### 2.4. Lifestyle Indices

Data on the patients’ current smoking habit, physical activity level, number of meals consumed per day, amount of sleep and whether or not they could sleep well were obtained from self-administered questionnaires conducted on the same day of anthropometric measurement.

### 2.5. Blood Collection, and Blood and Biochemical Tests

Patient blood was collected on the day or within 1 week of the anthropometric measurements and dietary and lifestyle survey. Data on the white blood cell count (WBC; cells µL^−1^), lymphocyte percentage (Lym%), albumin level (g dL^−1^), total cholesterol level (T-chol; mg dL^−1^), low-density lipoprotein cholesterol level (LDL-chol; mg dL^−1^), high-density lipoprotein cholesterol level (HDL-chol; mg dL^−1^), triglyceride level (TG; mg dL^−1^), fasting glucose level (mg dL^−1^), CD4 percentage (CD4%), CD8 percentage (CD8%) and HIV-RNA level were collected from the medical charts of the patients. Because the WBC, TLC, CD4 count and CD8 count are highly variable indices, we averaged the data from three different time points (the time of the survey and two previous outpatient visits). Data on the nadir CD4 count, duration of ART, current key drug, number of years since the HIV-positive diagnosis, history of a diagnosis of acquired immunodeficiency syndrome (AIDS) and comorbidities were also collected from the medical charts.

### 2.6. Arteriosclerosis-Related Indices

Blood pressure pulse wave tests were performed on a separate day from the outpatient clinic. The higher value of the left or right side of the branchial-ankle pulse wave velocity (baPWV), which is an index of arterial stiffness [16], was used in the analysis as baPWV max. The Ankle-Branchial Pressure Index (ABI), which is an indicator of arteriosclerosis obliterans [17], was set at the higher value on either the right or left sides. An ABI ≤ 0.9 is considered to indicate the presence of stenotic lesions in lower limb arteries with a specificity of almost 100%, and an ABI ≥ 1.4 is considered to indicate the presence of arterial calcification [18]. Therefore, these criteria were used as the cutoff values for evaluating the baPWV. These data were available for 97 of the 289 subjects in this study.

### 2.7. Evaluation of Metabolic Syndrome

Metabolic syndrome was defined as a waist circumference ≥ 85 cm (in males) and the presence of two or more of the following: abnormal lipid metabolism (TG ≥ 150 mg dL^−1^ and/or HDL-chol < 40 mg dL^−1^), hypertension (systolic blood pressure ≥ 130 mm Hg and/or diastolic blood pressure ≥ 85 mm Hg) and abnormal glucose metabolism (fasting blood glucose level ≥ 110 mg dL^−1^) [19]. A waist circumference ≥ 85 cm and the presence of one of the other factors were considered to indicate pre-metabolic syndrome.

### 2.8. Statistical Analysis

The normality of each index was confirmed based on the skewness (between −0.5 to 0.5) or *p* > 0.05 tested by Shapiro-Wilk’s W test. The Mann-Whitney U test, Chi-squared test and Fisher’s exact test were used to compare the differences between groups. For multiple logistic regression analysis, factors that showed a statistically significant difference between the subjects with a CD4 count of ≥500 cells (mm^3^)^−1^ and those with a CD4 count of <500 cells (mm^3^)^−1^ were used. A stepwise method (variable increase by the likelihood ratio) was used. IBM SPSS version 22.0 software (SPSS Inc., Chicago, IL, USA) was used for the analysis.

### 2.9. Ethical Considerations

Informed consent was obtained from all subjects involved in the study. This study was conducted with the approval of the Ethics Committee of the Osaka City General Medical Center (ID: 1312063) and Nara Medical University Hospital (ID: 1200).

## 3. Results

The patient characteristics are shown in Table 1. The CD4 ≥ 500 cells (mm^3^)^−1^ group was significantly younger and had a higher BMI than the CD4 < 500 cells (mm^3^)^−1^ group. In terms of HIV-related indices, the WBC, Lym%, TLC, CD4 count, CD8 count, CD4/CD8 ratio and nadir CD4 count were all significantly higher. The rate of having a history of an AIDS diagnosis was significantly lower, and the duration since the HIV-positive diagnosis and the duration of ART were significantly longer in the CD4 ≥ 500 cells (mm^3^)^−1^ group than in the CD4 < 500 cells (mm^3^)^−1^ group. The use of INSTIs was greater than 60% in both groups.

The blood biochemical and anthropometry measures are shown in Appendix A. The nutrient and food group intakes are shown in Appendix A, respectively. Data on the other lifestyle factors are shown in Appendix A.

For the indices that showed a statistically significant difference between the CD4 ≥ 500 cells (mm^3^)^−1^ and CD4 < 500 cells (mm^3^)^−1^ groups (Table 1 and Appendix A), the cutoff values for discriminating between the CD4 ≥ 500 cells (mm^3^)^−1^ and CD4 < 500 cells (mm^3^)^−1^ groups were determined from the receiver operating characteristic (ROC) curves (Table 2). For example, a BMI of 23.5 kg (m^2^)^−1^ or higher, a BF% of 25.1% or higher, 28.5 months or more of ART and a period of 5.5 years or more since the HIV-positive diagnosis were considered to be the cutoff values for discriminating between the two groups.

Based on the cutoff values obtained from the ROC curves, each index was transformed into a binary variable, and multiple logistic regression analysis was performed (Table 3). Factors that were significantly associated with a CD4 count of ≥500 cells (mm^3^)^−1^ included the age (≤45.5 years), BF% (≥25.1%), nadir CD4 count (≥137.5 cells (mm^3^)^−1^), duration of ART (≥28.5 months), years since the HIV-positive diagnosis (≥5.5 years) and cholesterol intake (≤265.3 mg 1000 kcal^−1^).

Since metabolic syndrome is a risk factor for atherosclerosis and cardiovascular disease, the cutoff values of BF% for metabolic syndrome and pre-metabolic syndrome were calculated using the ROC curves; the cutoff values were 25.5% for metabolic syndrome and 23.4% for pre-metabolic syndrome (Table 4). As surrogate markers for the risk of developing cardiovascular disease, we also determined from the ROC curves the cutoff values of BF% for a baPWV ≥ 1800 cm s^−1^, which corresponds to a high risk on the Framingham risk score, and for a baPWV ≥ 1400 cm s^−1^, which corresponds to a moderate risk on the Framingham risk score; the cutoff values were 25.8% for a baPWV ≥ 1800 cm s^−1^ and 24.0% for a baPWV ≥ 1400 cm s^−1^.

## 4. Discussion

In this study, the age, BF%, nadir CD4 count, duration of ART, years since the HIV-positive diagnosis and cholesterol intake showed a significant association with the CD4 count. The cutoff value of BF% for a CD4 count of ≥500 cells (mm^3^)^−1^ was 25.1% and over. The cutoff value of BF% for the lower risk of cardiovascular disease was 25.5% and lower. This suggests that 25% is the optimal BF% when considering the balance between CD4 count management and cardiovascular disease risk.

Multivariate analysis showed that an age ≤ 45.5 years, nadir CD4 count of ≥137.5 cells (mm^3^)^−1^ and ≥28.5 months of ART were significant factors for a CD4 count of ≥500 cells (mm^3^)^−1^. Montarroyos et al. reported that with ART, HIV-positive individuals >40 years of age showed a significantly smaller elevation in the CD4 count when compared to those ≤40 years of age [20]. Other studies also reported that age was independently associated with CD4 count recovery [21], and that older individuals took more time than younger individuals to recover the CD4 count after the initiation of ART [22]. Aging lowers a person’s adaptive immune system, making it harder for older patients to reach a CD4 count of 500 cells (mm^3^)^−1^. Lifson et al. reported that when ART was initiated at a nadir CD4 count of 125 cells (mm^3^)^−1^, it took about 5 years on average to reach a CD4 count of 500 cells (mm^3^)^−1^, and at a nadir CD4 count of 200 to 350 cells (mm^3^)^−1^, it took about 1.5 years for the CD4 count to reach 500 cells (mm^3^)^−1^ [23]. A lower nadir CD4 count means that there is more damage to CD4 T cells before ART initiation, and the results of the present study also support the notion that starting ART earlier leads to higher CD4 counts.

A notable new finding of this study was that a BF% of 25.1% or more was an independent factor for a CD4 count of ≥500 cells (mm^3^)^−1^. There are several reports on the association between BMI and the CD4 count. The rate of CD4 count recovery was high after the initiation of ART in individuals with a pre-treatment BMI of 25 kg (m^2^)^−1^ or more [24,25]. In a cross-sectional study, a BMI ≥ 25 kg (m^2^)^−1^ was associated with significantly higher CD4 counts [9]. Although the mechanism of the relationship between the BMI and CD4 count remains unclear, leptin has been suggested to be involved. Leptin is an adipocytokine, and its concentration in blood is greatly influenced by increases in BF mass. Leptin promotes the maturation of T cells from the thymus and prolongs the lifespan of T cells by slowing the rate of apoptosis [26]. Furthermore, leptin regulates the differentiation of CD4 and CD8 double-negative cells into CD4 and CD8 double-positive cells, and CD4 and CD8 double-positive cells into CD4-positive CD8-negative cells both in vitro and in vivo [27]. In the present study, both the BMI and BF% were entered into the multivariate analysis, and the BF%, but not the BMI, was extracted as a significant factor. This result provides strong support for the association between adipocytes, leptin and the CD4 count.

Although the multivariate analysis showed that a BF% of 25.1% or higher is desirable for a CD4 count of ≥500 cells (mm^3^)^−1^, an increase in the BF% is reported to be associated with cardiovascular disease risk [12]. The increase in cardiovascular disease risk cannot be ignored in HIV-positive individuals since they have chronic inflammation due to the human retrovirus [28], and it is reported to increase the risk for cardiovascular disease [29,30]. In this study, the cutoff value of the BF% was 25% to 26% for both metabolic syndrome and the baPWV. Hiroshige et al. reported that the average BF% of 267 working Japanese males who were negative for HIV was 22.2% [31]; our cutoff values for CD4 count management and cardiovascular disease risk (25.1% and 25.5%, respectively) were both slightly higher. In addition, Asians tend to have a higher BF% than Caucasians at the same BMI value [32,33]. Thus, the body composition is the important factor for the Japanese population. Moreover, HIV individuals with ART present lipodystrophy (lipoatrophy/lipohypertrophy). Soares et al. reported that anthropometric measurement is needed to evaluate the nutritional status of HIV individuals [34]. This also supports that body composition measurement is essential for HIV individuals.

This study has a few limitations. This study included only male patients, and thus, the results cannot be applied to female HIV patients. However, according to the AIDS Outbreak Trends Survey, more than 90% of the HIV cases in Japan are male. In addition, new HIV infections are generally reported in patients in their 20 s to 40 s, and new AIDS cases are generally reported in patients in their 40 s [35]. Therefore, the population of this study is considered closely to reflect the distribution of the population of people living with HIV in Japan. The cross-sectional nature of the data is another limitation of this study. Of the factors included in the analysis, the nadir CD4 count and the duration of ART were retrospectively collected information taken from the data records of the patients. This made it possible to state a causal relationship even with the cross-sectional data. However, the other factors extracted by multivariate analysis can only be described as relevant in this study. In the future, a prospective cohort study should be performed to examine whether the BF% affects the control of the CD4 count.

## *5.* Conclusions

This study showed that the BF%, and not the BMI, was associated with the CD4 count, and that a BF% of 25.1% or more may be advantageous for maintaining the CD4 count in HIV-positive individuals. However, a BF% of 25.5% or more may increase the risk of coronary artery disease. Thus, for the management of HIV-positive individuals, regular measurements of body composition are recommended.

## Figures and Tables

**Figure 1 nutrients-14-00428-f001:**
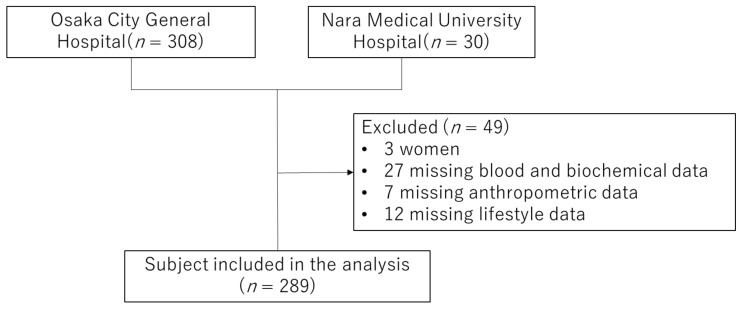
Flow chart for the study subject.

**Table 1 nutrients-14-00428-t001:** Characteristics of the total patients and separated by CD4 count 500 cells (mm^3^)^−1^.

		Total (*n* = 289)		CD4 ≥ 500 cells (mm^3^)^−1^ (*n* = 160)		CD4 < 500 cells (mm^3^)^−1^ (*n* = 129)		*p* *
		Median	(25th, 75th Percentile) or (%)	Median	(25th, 75th Percentile) or (%)	Median	(25th, 75th Percentile) or (%)	
Age (years old)		45	(39, 54)	44	(37, 51)	47	(40, 59)	0.011
Height (cm)		170.0	(166.0, 173.3)	170.0	(165.3, 174.0)	170.0	(166.0, 173.0)	0.802
Weight (kg)		67.2	(60.0, 75.3)	68.3	(61.0, 76.7)	65.4	(58.4, 72.9)	0.029
BMI (kg (m^2^)^−1^)		23.3	(21.0, 25.8)	23.7	(21.4, 26.5)	22.7	(20.6, 25.0)	0.010
WBC		5566	(4847, 6612)	6088	(5316, 7285)	5143	(4575, 5917)	<0.0001
Lymp (%)		34.6	(29.4, 39.7)	36.9	(31.1, 41.0)	31.3	(26.3, 37.0)	<0.0001
TLC (cells (mm^3^)^−1^)		1880	(1491, 2408)	2242	(1787, 2618)	1554	(1283, 1849)	<0.0001
CD4 count (cells (mm^3^)^−1^))		532	(397, 667)	656	(578, 764)	378	(296, 449)	<0.0001
CD8 count (cells (mm^3^)^−1^))		830	(632, 1132)	939	(699, 1210)	720	(533, 939)	<0.0001
CD4/CD8 ratio		0.64	(0.46, 0.86)	0.74	(0.57, 0.97)	0.50	(0.35, 0.67)	<0.0001
Nadir CD4 count (cells (mm^3^)^−1^))		133	(47, 213)	175	(100, 254)	71	(28, 140)	<0.0001
HIV-RNA ≥ 20 (copies mL^−1^)		6	(2.1)	3	(1.9)	3	(2.3)	0.552
History of AIDS diagnosis		103	(35.6)	41	(25.6)	62	(48.1)	<0.0001
Duration since HIV diagnosis (years)		8	(4, 11)	8	(5, 12)	6	(3, 11)	0.001
Duration of ART (months)		65	(35, 108)	68.5	(41, 118)	58	(28, 103)	0.033
Key drug								0.196
	NNRTIs	45	(15.6)	26	(16.3)	19	(14.7)	
	PIs	58	(20.1)	26	(16.3)	32	(24.8)	
	INSTIs	186	(64.4)	108	(67.5)	78	(60.5)	
Comorbidities								
	Diabetes	29	(10.1)	14	(8.8)	15	(11.7)	0.405
	HTN	46	(16.0)	24	(15.0)	22	(17.2)	0.615
	Hyperlipidemia	55	(19.1)	37	(23.1)	18	(14.1)	0.052
	Fatty liver	28	(9.7)	20	(12.5)	8	(6.3)	0.075
	Virus hepatitis	75	(26.0)	33	(20.6)	42	(32.6)	0.021
Metabolic syndrome		90	(30.1)	53	(33.1)	37	(28.7)	0.445
Pre metabolic syndrome		38	(13.1)	23	(14.4)	15	(11.6)	0.600

*p**: CD4 ≥ 500 cells (mm^3^)^−1^ vs. CD4 < 500 cells (mm^3^)^−1^; Mann-Whitney U test for continuous variables, x2 test or Fisher’s exact test for other variables Abbreviations and symbol: BMI, body mass index; WBC, white blood cell; Lymp, lymphocyte; TLC, total lymphocyte count; CD4, CD4 positive lymphocyte; CD8, CD8 positive lymphocyte; ART, Antiretroviral therapy; NNRTIs, Non-nucleoside reverse transcriptase inhibitors; PI, protease inhibitors; INSTI, Integrase strand transfer inhibitors; HTN, hypertension.

**Table 2 nutrients-14-00428-t002:** Cutoff values for CD4 ≥ 500 cells (mm^3^)^−1^ by ROC curve.

	AUC	*p*	Cutoff Value
Age (years old)	0.411	0.011	≤45.5
BMI (kg (m^2^)^−1^)	0.598	0.005	≥23.5
Waist circumference (cm)	0.58	0.024	≥87.1
BF%	0.591	0.01	≥25.1
SMI (kg (m^2^)^−1^)	0.585	0.016	≥7.7
Alb (g dL^−1^)	0.583	0.018	≥4.25
LDL-chol (mg dL^−1^)	0.589	0.011	≥130.5
non-HDL (mg dL^−1^)	0.572	0.041	≥152.3
Nadir CD4 count (cells (mm^3^)^−1^)	0.74	<0.0001	≥137.5
Duration of ART (months)	0.573	0.04	≥28.5
Duration since HIV diagnosis (years)	0.621	0.001	≥5.5
NPC-N ratio	0.577	0.03	≥134
Calcium intake (mg 1000 kcal^−1^)	0.424	0.032	≤167.8
Potassium intake (mg 1000 kcal^−1^)	0.424	0.03	≤601.9
Zinc intake (mg 1000 kcal^−1^)	0.423	0.028	≤4.44
Pantothenic acid intake (mg 1000 kcal^−1^)	0.429	0.043	≤3.47
Cholesterol intake (mg 1000 kcal^−1^)	0.427	0.04	≤265.3
Weight of beans intake (mg 1000 kcal^−1^)	0.377	<0.0001	≤16.4

Abbreviations and symbol: AUC, area under the curve; BMI, body mass index; BF%, body fat percentage; SMI, skeletal muscle mass index; Alb, albumin; LDL-chol, low-density lipoprotein cholesterol; non-HDL, non-high-density lipoprotein cholesterol; CD4, CD4 positive lymphocyte; ART, Antiretroviral therapy; NPC-N ratio, non-protein calorie-nitrogen ratio.

**Table 3 nutrients-14-00428-t003:** Univariate and multivariate logistic regression analysis for CD4 ≥ 500 cells (mm^3^)^−1^.

	Univariate			Multivariate		
	OR	95% CI	*p*	OR	95% CI	*p*
Age ≤ 45.5	2.20	(1.35, 3.59)	0.002	2.62	(1.30, 5.27)	0.007
BMI ≥ 23.5 kg (m^2^)^−1^	1.98	(1.21, 3.23)	0.007			
Waist circumference ≥ 87.1 cm	2.09	(1.27, 3.46)	0.004			
BF% ≥ 25.1 %	2.04	(1.24, 3.36)	0.005	3.43	(1.71, 6.88)	0.001
SMI ≥ 7.7 kg (m^2^)^−1^	1.91	(1.16, 3.15)	0.011			
Alb ≥ 4.25 g dL^−1^	2.07	(1.25, 3.44)	0.005			
LDL-chol ≥ 130.5 mg dL^−1^	2.02	(1.15, 3.55)	0.014			
non-HDL ≥ 152.3 mg dL^−1^	2.08	(1.25, 3.45)	0.005			
Nadir CD4 count ≥ 137.5 (mm^3^)^−1^	6.82	(3.97, 11.73)	<0.0001	6.94	(3.50, 13.75)	<0.0001
Duration of ART ≥ 28.5 months	2.93	(1.52, 5.66)	0.001	2.80	(1.04, 7.50)	0.041
Duration since HIV diagnosis ≥ 5.5 years	2.89	(1.73, 4.81)	<0.0001	5.06	(2.35, 10.90)	<0.0001
NPC-N ratio ≥ 134	2.62	(1.50, 4.57)	0.001			
Calcium intake ≤ 167.8 mg 1000 kcal^−1^	2.21	(1.18, 4.15)	0.014			
Potassium intake ≤ 601.9 mg 1000 kcal^−1^	2.90	(1.61, 5.22)	<0.0001			
Zinc intake ≤ 4.44 mg 1000 kcal^−1^	2.84	(1.62, 4.97)	<0.0001			
Pantothenic acid intake ≤ 3.47 mg 1000 kcal^−1^	2.80	(1.64, 4.78)	<0.0001			
Cholesterol intake ≤ 265.3 mg 1000 kcal^−1^	3.38	(1.63, 6.97)	0.001	3.37	(1.23, 9.18)	0.018
Pulses intake ≤ 16.35 mg 1000 kcal^−1^	2.41	(1.47, 3.95)	<0.0001			
No history of AIDS diagnosis	0.40	(0.24, 0.66)	<0.0001			
No diagnosis of virus hepatitis	0.61	(0.35, 1.06)	0.078	0.39	(0.19, 0.82)	0.013
No current smoking habit	2.09	(1.26, 3.49)	0.005			
Meals per day ≥ 3 times	0.53	(0.32, 0.88)	0.014			

Abbreviations and symbol: BMI, body mass index; BF%, body fat percentage; SMI, skeletal muscle mass index; Alb, albumin; LDL-chol, low-density lipoprotein cholesterol; non-HDL, non-high-density lipoprotein cholesterol; CD4, CD4 positive lymphocyte; ART, Antiretroviral therapy; NPC-N ratio, non-protein calorie-nitrogen ratio.

**Table 4 nutrients-14-00428-t004:** Cutoff values of body fat percentage for arteriosclerosis related indices.

	AUC	*p*	BF % Cutoff Values
Metabolic syndrome	0.800	<0.0001	25.5%
Metabolic syndrome and preliminary metabolic syndrome	0.837	<0.0001	23.4%
baPWV ≥ 1800 cm s^−1^	0.647	0.083	25.8%
baPWV ≥ 1400 cm s^−1^	0.659	0.011	24.0%

Abbreviations and symbol: AUC, area under the curve; BF%, body fat percentage; baPWV, branchial-ankle pulse wave velocity.

## Data Availability

Not applicable.

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
