# Peer review of "The Body Fat Percentage Rather Than the BMI Is Associated with the CD4 Count among HIV Positive Japanese Individuals"

_nutrients, 2022, doi:10.3390/nu14030428_

Round 1

Reviewer 1 Report

 # Review

  • Lines 141-142: The following statement is not in line with the results in Table 1: „the CD8 count was significantly lower in the C CD4≥ 141 500/mm3 group than in the CD4.“
  • Lines 195-197: the following reference is not correct: „Lifson et al. reported that when ART was initiated at a nadir CD4 count of 125 cells/mm3, it took about 5 years on average to reach a CD4 count of 500 cells/mm3, and at a nadir CD4 count of 200 to 350 cells/mm3, it took about 1.5 years for the CD4 count to reach 500 cells/mm3 [23]“. Kindly re-check reference 23.
  • English editing for some minor corrections is recommended.

Author Response

I very much appreciate your through review. These are point-by-point response to your comments. 

  • Lines 141-142: The following statement is not in line with the results in Table 1: „the CD8 count was significantly lower in the C CD4≥ 141 500/mm3 group than in the CD4.“

>Thank you for the correction. I added CD8 count in the previous line. CD8 was significantly higher in the CD4 ≥ 500/mm3 group than CD4 < 500/mm3 group.

  • Lines 195-197: the following reference is not correct: „Lifson et al. reported that when ART was initiated at a nadir CD4 count of 125 cells/mm3, it took about 5 years on average to reach a CD4 count of 500 cells/mm3, and at a nadir CD4 count of 200 to 350 cells/mm3, it took about 1.5 years for the CD4 count to reach 500 cells/mm3 [23]“. Kindly re-check reference 23.

>Thank you for the correction. I mistakenly inserted the wrong reference. I reinserted corrected one.

  • English editing for some minor corrections is recommended.

>Thank you for the comment. I reviewed whole manuscript.

Reviewer 2 Report

The manuscript titled “The Body Fat Percentage rather than the BMI is Associated With the CD4 Count Among HIV Positive Japanese Individuals” by Shoji et al is well written. However, my major concern is on the association between BF% and cardiovascular disease.

Line 33-34 and 184: Authors stated the BF% <25.5% is associated with cardiovascular disease risk. I think it should be other way around. Because body fat has consistently been associated with an increased risk for metabolic diseases and cardiovascular disease. An increase in body fat can directly contribute to heart disease through atrial enlargement, ventricular enlargement, and atherosclerosis. However, in conclusion, Authors stated “a BF% of 25.5% or more may increase the risk of coronary artery disease”.

Minor comments

Provide reference for “There have been several reports on the association between BMI and the CD4 count”.

Data for multivariate logistics regression is missing in Table 3/ Table 3 is partial.

There are few spelling mistakes here and there. For example, abbrebiation and mascle.

Author Response

I very much appreciate your through review. These are point-by-point response to your comments. 

  • Line 33-34 and 184: Authors stated the BF% <25.5% is associated with cardiovascular disease risk. I think it should be other way around. Because body fat has consistently been associated with an increased risk for metabolic diseases and cardiovascular disease. An increase in body fat can directly contribute to heart disease through atrial enlargement, ventricular enlargement, and atherosclerosis. However, in conclusion, Authors stated “a BF% of 25.5% or more may increase the risk of coronary artery disease”.

>Thank you for the concern. My expression regarding cardiovascular disease risk was opposite against what I wanted to tell. I added “lower” in front of cardiovascular disease risk, so the sentence became “The cutoff value of BF% for a CD4≥500/mm3 and lower cardiovascular disease risk were ≥25.1%, and ≤25.5%, respectively.”

  • Provide reference for “There have been several reports on the association between BMI and the CD4 count”.

>Thank you for the concern. The following sentences are explaining references. For example, reference [9]&[10] in the introduction(line56-58), [9],[24]&[25] in the discussion(line221-224).

  • Data for multivariate logistics regression is missing in Table 3/ Table 3 is partial.

>I apologize that I inserted tables inappropriately in the manuscript. I reinserted tables.

  • There are few spelling mistakes here and there. For example, abbrebiation and mascle.

>Thank you for the correction. I corrected spelling.

Reviewer 3 Report

Reviewer report

Researchers presented interesting results on the association between body fat % and CD4 count in HIV-1 infected individuals treated with ART for >12 months.

Major concerns;

  1. There is no description on the duration of HAART in the Table 1. Researchers presented interesting results.

...

Researchers presented an interesting result.

...

  1. Authors’d better to cite following paper in the discussion because they have published the importance of body fat percentage.
  • Discordance between body mass index and anthropometric measurements among HIV-1-infected patients on antiretroviral therapy and with lipoatrophy/lipohypertrophy syndrome. Rev Inst Med Trop Sao Paulo. 2015;57(2):105-10.

Minor points;

  1. In Table 1, Key Drug -> Key drug, Pis -> PIs.
  2. Please use “≥” in the text and tables 1, 2, 3, and 4 (not > plus =).
  3. In Table 2, please insert blank mg/1000kcalà mg/1000 kcal, BF% (%) -> BF%
  4. Please insert blank at CD4≥500/mm3-à CD4 ≥ 500/mm3.
  5. Line 59, it is remains->it remains….
  6. Lines 61 and 67, among body weight, body composition and the CD4 count in Japanese…
  7. Please delete lines 100 and 101.
  8. Line 141: There is mismatch between text and with Table 1 and delete C CD4 and all % in Table 1.
  9. Line 155, 23.53 kg/m2à5 kg/m2

Author Response

I very much appreciate your through review. These are point-by-point response to your comments. 

  1. There is no description on the duration of HAART in the Table 1. Researchers presented interesting results.

>Thank you for the concern. “Duration of HIV medication(months)” was the duration of HAART. I changed to “Duration of ART(months)”.

  1. Authors’d better to cite following paper in the discussion because they have published the importance of body fat percentage.
  • “ Discordance between body mass index and anthropometric measurements among HIV-1-infected patients on antiretroviral therapy and with lipoatrophy/lipohypertrophy syndrome. Rev Inst Med Trop Sao Paulo. 2015;57(2):105-10.

>Thank you for the suggestion. I added in the 4th paragraph of discussion (line245-248).

Minor points;

  1. In Table 1, Key Drug -> Key drug, Pis -> PIs. 
  2. Please use “≥” in the text and tables 1, 2, 3, and 4 (not > plus =).
  3. In Table 2, please insert blank mg/1000kcalà mg/1000 kcal, BF% (%) -> BF% 
  4. Please insert blank at CD4≥500/mm3-à CD4 ≥ 500/mm3. 
  5. Line 59, it is remains->it remains….
  6. Lines 61 and 67, among body weight, body composition and the CD4 count in Japanese… 
  7. Please delete lines 100 and 101.
  8. Line 141: There is mismatch between text and with Table 1 and delete C CD4 and all % in Table 1. 
  9. Line 155, 23.53 kg/m2à5 kg/m

>I really appreciate the correction. I corrected point 1-9.